TRANSPARENT PROCESS OPEN ACCESS Life Science Alliance
# A C/EBPα–Wnt connection in gut homeostasis and carcinogenesis

Julian Heuberger[1,*], Undine Hill[1,*], Susann Förster[1], Karin Zimmermann[1], Victoria Malchin[1], Anja A Kühl[4], Ulrike Stein[2,3], Michael Vieth[5], Walter Birchmeier[1], Achim Leutz[1,6]

**We explored the connection between C/EBPα (CCAAT/enhancer-binding protein α) and Wnt signaling in gut homeostasis and carcinogenesis. C/EBPα was expressed in human and murine intestinal epithelia in the transit-amplifying region of the crypts and was absent in intestinal stem cells and Paneth cells with activated Wnt signaling. In human colorectal cancer and murine APC$^{Min/+}$ polyps, C/EBPα was absent in the nuclear β-catenin–positive tumor cells. In chemically induced intestinal carcinogenesis, C/EBPα KO in murine gut epithelia increased tumor volume. C/EBPα deletion extended the S-phase cell zone in intestinal organoids and activated typical proliferation gene expression signatures, including that of Wnt target genes. Genetic activation of β-catenin in organoids attenuated C/EBPα expression, and ectopic C/EBPα expression in HCT116 cells abrogated proliferation. C/EBPα expression accompanied differentiation of the colon cancer cell line Caco-2, whereas β-catenin stabilization suppressed C/EBPα. These data suggest homeostatic and oncogenic suppressor functions of C/EBPα in the gut by restricting Wnt signaling.**


## Introduction

The Wnt signaling pathway is activated in more than 80% of colorectal cancer (CRC) cases, mostly produced by mutations of the tumor suppressor gene *APC* (adenomatous polyposis coli). *APC* loss prevents the degradation of β-catenin, the intracellular mediator of Wnt signaling, and results in enhanced β-catenin translocation into the nucleus and subsequent activation of the Wnt target genes that promote proliferation (Fearon & Vogelstein, 1990; Sieber et al, 2000; Fodde & Smits, 2001; McCart et al, 2008; Kwong & Dove, 2009).

Cell differentiation induced by the transcription factor C/EBPα (CCAAT/enhancer-binding protein α) is negatively correlated with canonical Wnt signaling (Kang et al, 2007; Kawai et al, 2007). In an adipogenic precursor cell line, activated Wnt signaling prevented C/EBPα-induced differentiation (Kawai et al, 2007). Wnt signaling activation with recombinant Wnt3a or the glycogen synthase kinase 3β (GSK3β) inhibitor CHIR99021 in stromal progenitor ST2 cells reduced C/EBPα expression (Kang et al, 2007) and caused a shift from adipogenic to osteoblastic cell fate, whereas transgenic re-expression of C/EBPα rescued the adipogenic differentiation program (Kawai et al, 2007). In the HL7702 hepatic cell line, transgenic β-catenin expression repressed endogenous C/EBPα expression (Wang et al, 2013), suggesting that the antagonism of C/EBPα and Wnt signaling might represent a more general mechanism in proliferation versus differentiation control and raises the possibility of an oncogene/tumor suppressor relationship.

Although C/EBPα expression was previously detected in the intestinal epithelium, little is known about C/EBPα-dependent proliferation control or tumor suppressor functions in the gut and its relationship to canonical Wnt signaling (Oesterreicher et al, 1998; Silviera et al, 2012). In the present study, we combined the histopathological analysis of human colon cancer with experimental chemical tumorigenesis, conditional murine genetics in organoid cultures, and cell biological data to explore the role of a connection between Wnt signaling and C/EBPα in the gut. Our data reveal C/EBPα and canonical Wnt signaling as opponents in epithelial growth control and suggest a tumor suppressor function of C/EBPα in the mammalian gut.

## Results

### C/EBPα expression in normal intestinal epithelia and CRC tissue

To address C/EBPα function and its relationship with Wnt signaling in colorectal carcinogenesis, we examined normal and cancerous human colon tissues by immunohistochemistry (IHC) (Fig 1). The samples comprised biopsies of normal epithelium (n = 18),

[1]Max Delbrück Center for Molecular Medicine, Berlin, Germany [2]Experimental and Clinical Research Center, Charite—Universitätsmedizin Berlin and Max Delbrück Center for Molecular Medicine, Berlin, Germany [3]German Cancer Consortium (Deutsches Konsortium für Translationale Krebsforschung), Heidelberg, Germany [4]Charité—Universitätsmedizin Berlin, Corporate Member of Freie Universität, Humboldt-Universität zu Berlin, and Berlin Institute of Health, Berlin, Germany [5]Klinikum Bayreuth, Institute for Pathology, Bayreuth, Germany [6]Institute of Biology, Humboldt University of Berlin, Berlin, Germany

Correspondence: aleutz@mdc-berlin.de
*Julian Heuberger and Undine Hill shared first authorship

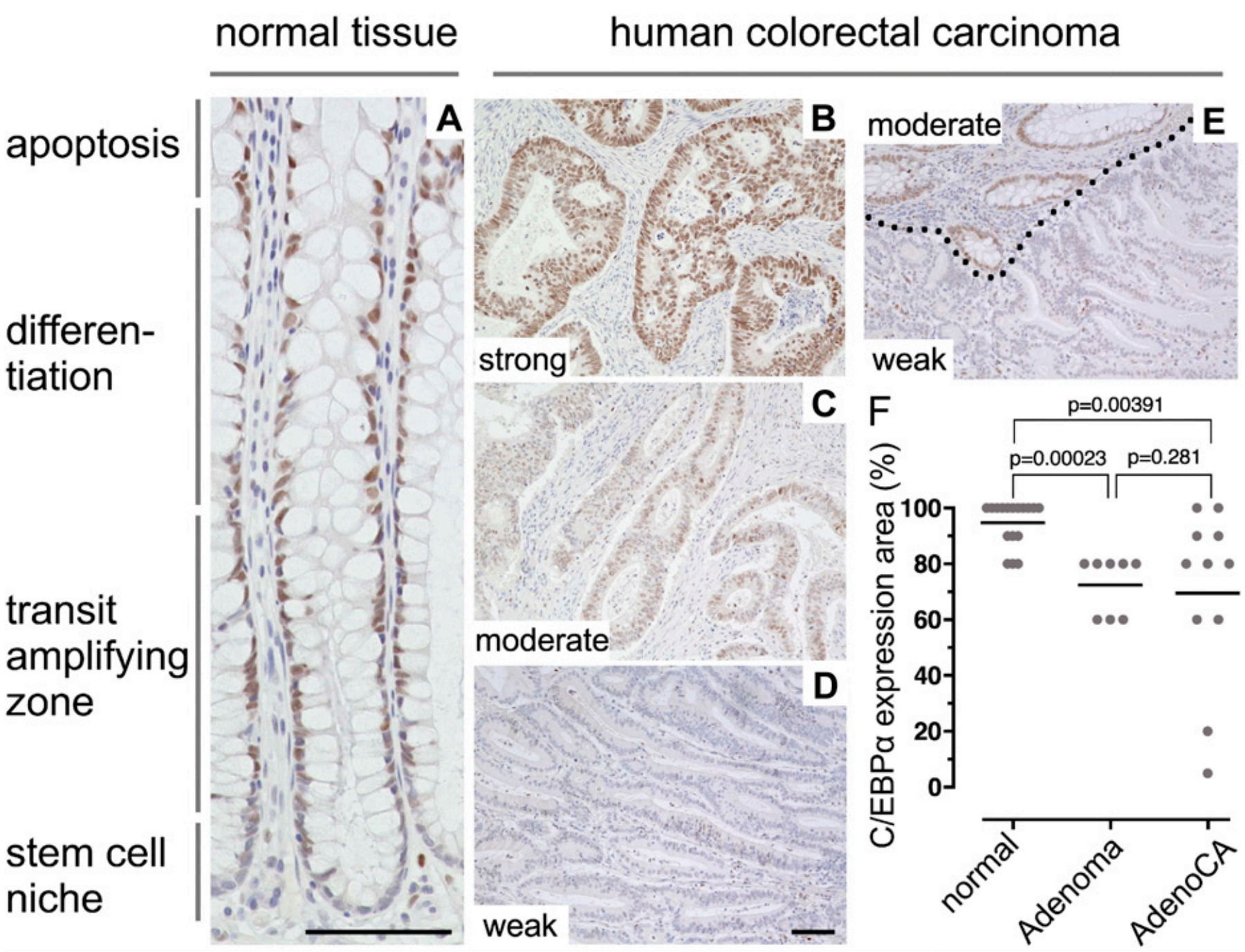


**Figure 1. C/EBPα expression in the normal human colon and colorectal carcinoma.**
**(A)** C/EBPα IHC on paraffin sections of healthy human colon. C/EBPα is expressed in the nuclei of colonic crypt cells in the TA zone; there is low or no expression in cells at the crypt base. **(B–D)** C/EBPα IHC on paraffin sections of human colorectal adenocarcinoma biopsies with different C/EBPα expression levels as indicated. **(E)** Border between healthy tissue (moderate C/EBPα expression) and adjacent cancerous tissue (dotted line, weak C/EBPα expression). **(F)** Quantification of C/EBPα-expressing areas in normal tissue, adenoma (low-grade intraepithelial neoplasia/dysplasia), and adenocarcinomas (Adeno CA) as indicated. Mann–Whitney test, *P*-values above; data are listed in Table S1. Scale bars indicated in (A) and (D): 100 μm.

spontaneous colorectal adenoma (n = 8), and spontaneous colorectal adenocarcinoma (n = 11). In the normal human colon epithelium, C/EBPα was expressed in the nuclei in the transient proliferation zone and differentiated cells, but was largely absent in cells at the base of the crypts (Fig 1A). Histopathological evaluation of biopsies was measured as the percentage of the C/EBPα-positive area, and expression intensity of C/EBPα was scored as negative (0), weak (1), moderate (2), or strong (3) (Table S1).

Adenomas and adenocarcinomas versus normal epithelium showed differential expression of C/EBPα (Fig 1B–D and Table S1). Expression intensities varied in adenoma and carcinoma from strong to weak, with a trend toward reduced C/EBPα expression levels in adenoma and carcinoma (Table S1). Reduction in C/EBPα expression in the neoplasm versus adjacent non-neoplastic is demonstrated in Fig 1E (dotted line indicates the border of cancerous tissue). There was a pronounced reduction in the area of C/EBPα expression. In the normal colon epithelium, C/EBPα was expressed in 80–100% of the analyzed area. In contrast, in adenomas the C/EBPα expression region was reduced to 60–80% of the neoplastic compartment. In adenocarcinomas, C/EBPα expression areas ranged from 100% down to 5% of cancerous lesions. Overall, the expression areas were significantly reduced in adenocarcinoma compared to normal epithelium (Fig 1F). To address whether the diversity in C/EBPα expression involves Wnt signaling activity, we examined C/EBPα and nuclear β-catenin expression by immunofluorescence (IF) in colorectal adenoma and adenocarcinoma. In both, C/EBPα expression was observed in distinct areas of the neoplastic compartments that expressed low levels of β-catenin. In contrast, C/EBPα expression strongly decreased or was absent in cells with high nuclear β-catenin expression (Fig S1). These findings support the idea that activated Wnt signaling and C/EBPα expression in gut cells are mutually exclusive.

We first examined the status of activated Wnt signaling and C/EBPα expression in mouse gut to address their relationship in a model system that is amenable to experimental oncogenesis and targeted genetics. Immunostaining of sections of the small intestine of 15-wk-old C57BL/6 and Lgr5 reporter mice showed C/EBPα expression in transit-amplifying (TA) cells in the crypt (Fig 2A). C/EBPα was weakly expressed in Lgr5-positive stem cells at the bottom of the crypts (Fig 2A, arrowheads), but was absent in lysozyme-positive Paneth cells and terminally differentiated cells of the villus. Expression levels in the crypt were quantified from IF images comparing Lgr5-stem cells and other crypt cells of the region 1 to +5 cell, 6 to +8 cell, and 9 to +12 cells (Fig S2). In vivo EdU (5-ethynyl-2′-deoxyuridine) labelling of S-phase cells confirmed that C/EBPα expression was present in proliferating TA cells. However, cells labeled with EdU had the lowest C/EBPα expression, implying that C/EBPα-positive cells enter S-phase less frequently (Fig 2B). It therefore appears that C/EBPα is expressed in cells committed to differentiation and may restrict proliferation in the TA zone.

### C/EBPα expression is decreased in $APC^{Min/+}$ adenoma

$APC^{Min/+}$ mice develop intestinal polyps and adenomas because of a deficient β-catenin destruction complex that causes β-catenin stabilization (Su et al, 1993). We used $APC^{Min/+}$ mice to examine whether oncogenic activation of Wnt signaling decreased C/EBPα expression. There was enhanced β-catenin expression in the polyp cells, and in particular, in cells in the invading adenomatous tissue, but not in the adjacent normal/healthy tissue with differentiated goblet cells (Fig 3A), reminiscent of that observed in human colon cancer (Figs 1 and S1). Serial sections revealed strongly reduced C/EBPα expression in the adenomatous tissue, in particular at the basal areas of polyps that had the highest levels of nuclear

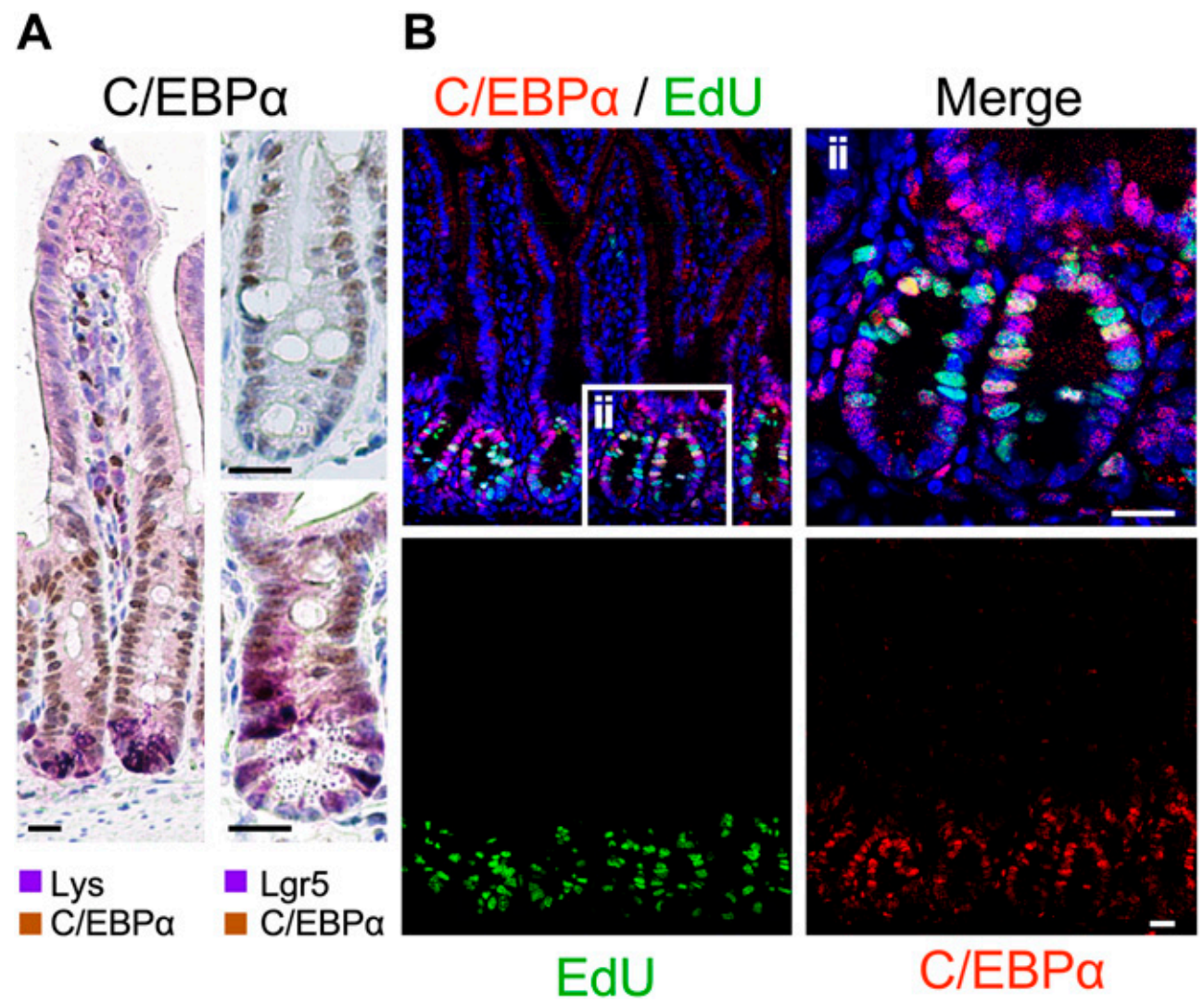


**Figure 2. C/EBPα expression in TA cells of the small intestinal crypt in mice.**
**(A)** C/EBPα IHC on paraffin sections; (left) double IHC of C/EBPα (brown) with lysozyme (Paneth cell marker, purple); (right) IHC of C/EBPα (brown) and Lgr5-GFP (purple) of Lgr5 reporter mice. (Top right) C/EBPα IHC. C/EBPα is not expressed in the high-Wnt Lgr5 stem cells and Paneth cells. C/EBPα expression is restricted to the TA cells. **(B)** Double IF staining of C/EBPα (red) and EdU-labeled S-phase cells (green); (ii) inset, as shown on the right with higher magnification. Scale bars: 20 μm.

β-catenin (Fig 3, inset, right of dotted line). However, adjacent to the adenomatous tissue C/EBPα expression was detected in normal cells that had lower levels of nuclear β-catenin (Fig 3, inset, left of the dotted line). Quantitative RT–PCR (qRT–PCR) of micro-dissected adenoma tissue and the neighboring healthy/normal intestinal tissue of APC$^{Min/+}$ mice confirmed 75% reduction in C/EBPα expression in adenomatous tissue with elevated Wnt signaling (Fig 3C). Double IF staining confirmed mutually exclusive expression of C/EBPα and β-catenin (Fig S3, upper panel). In addition, Ki67-positive proliferative cells in the cancerous lesions did not express C/EBPα (Fig S3, lower panel). These data show that Wnt-activated and proliferative cells in tumor lesions in both humans and mice do not express C/EBPα.

### Wnt signaling down-regulates C/EBPα in intestinal organoids

To assess whether Wnt/β-catenin signaling down-regulates C/EBPα expression, we examined the intestinal organoids from β-CatEx3$^{flox/+}$-Villin-Cre$^{ERT2}$ mice, which express stabilized gain-of-function (GOF) β-catenin after 4-OHT (4-hydroxytamoxifen)–induced Cre-mediated recombination. Organoids with elevated β-catenin exhibited an increase in Wnt target gene expression after the induction of recombination, as determined by qRT–PCR for Axin2 and the Wnt-dependent stem cell marker Lgr5 (Fig 4A). C/EBPα expression was severely reduced in GOF β-catenin organoids, as assessed by histological staining (Fig 4B) and after the induction of recombination protein blotting (Fig 4C). Collectively, the data from the APC$^{Min/+}$ mice and β-catenin GOF organoids showed that increased Wnt signaling reduces C/EBPα expression and presents the possibility that reduced C/EBPα expression may permit tumor progression.

### C/EBPα restricts tumor growth in murine colitis-associated cancer

To explore the function of C/EBPα in tumor progression, mice with conditional loss-of-function alleles of C/EBPα (C/EBPα$^{Flox/Flox}$-VilinCre$^{ERT2}$) were compared to controls (C/EBPα$^{Flox/Flox}$) in a chemically induced intestinal azoxymethane–dextran sodium sulfate (AOM-DSS) colitis-associated carcinogenesis model (Bollrath et al, 2009). After tamoxifen-induced C/EBPα depletion, tumorigenesis was induced by exposure to the colonotropic mutagen AOM and subsequent administration of the luminal toxin DSS. AOM causes β-catenin stabilization and nuclear translocation by inducing missense mutations in exon 3 of β-catenin (Greten et al, 2004). Fifteen weeks after AOM-DSS treatment, all mice developed on average 10 colitis-associated low-grade dysplasia in the distal colon, mostly confined to the mucosa and in some cases focal submucosal invasion with mild mucosal or partial minimal submucosal invasion. C/EBPα was entirely depleted in the dysplasias of the conditional

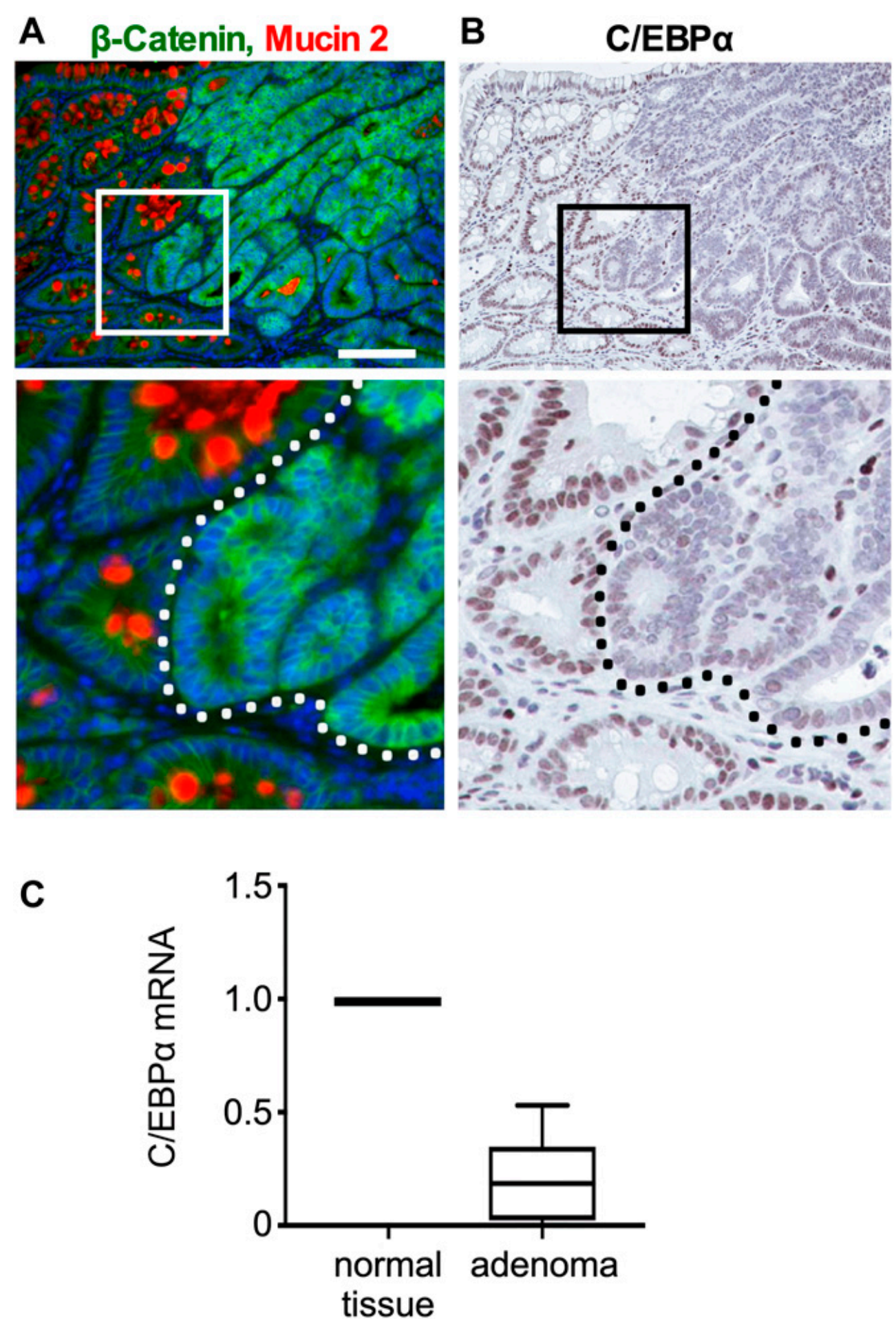


**Figure 3. Low C/EBPα expression in adenoma of APC$^{Min/+}$ mice with high β-catenin levels.**
**(A)** IF of β-catenin (green) and MUC2 (red, goblet cell marker) in adenoma sections of APC$^{Min/+}$ mice (scale bar: 100 μm). **(B)** IHC of C/EBPα (brown) on consecutive sections to (A). C/EBPα expression is greatly reduced in adenoma with high β-catenin levels. **(C)** Relative *Cebpa* mRNA expression in micro-dissected adenoma compared to normal surrounding intestinal tissue (n = 12).

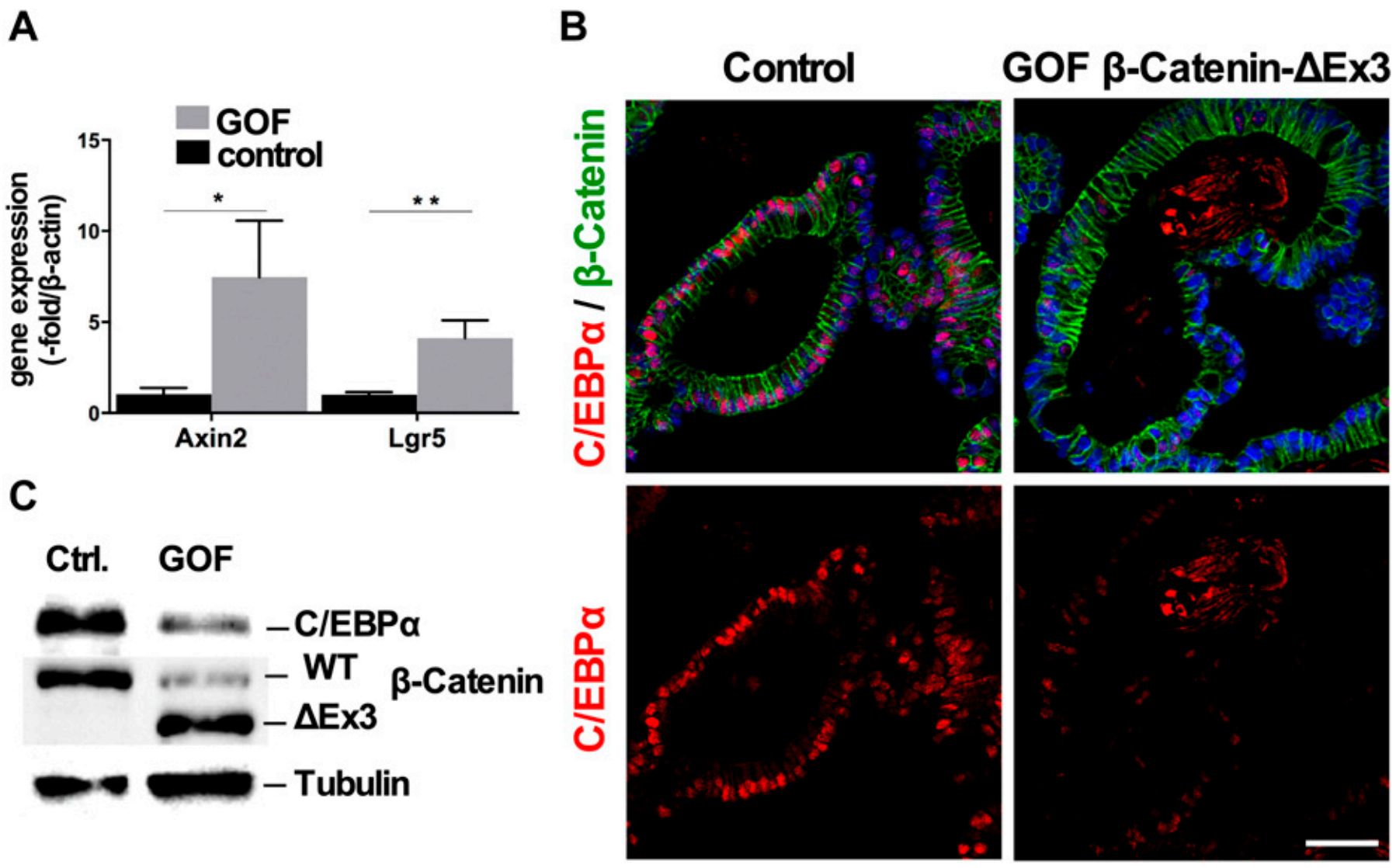


**Figure 4. Intestinal organoids with increased Wnt signaling have reduced C/EBPα expression.**
β-CatEx3$^{flox/+}$-Villin-Cre$^{ERT2}$ small intestinal organoid culture. β-Catenin stabilization was induced by single-day administration of 4-OHT (800 nM) in culture. **(A)** qRT–PCR comparing Wnt target gene expression in control and GOF β-catenin organoids (n = 3, unpaired *t* test, two-tailed, Axin2: **P* = 0.0234; Lgr5: ***P* = 0.0054). **(B)** IF of β-catenin (green) and C/EBPα (red) in control and GOF β-catenin small intestinal organoids. **(C)** Western blot analysis of control and GOF β-catenin protein lysates probed for C/EBPα and β-catenin; loading control is tubulin.

mutants; however, the control adenomatous lesions likewise had reduced C/EBPα expression (Fig 5A). Remarkably, dysplasia with conditional loss of C/EBPα had significantly increased size in the distal part of the colon, while overall numbers of adenomatous lesions remain unchanged (Fig 5B). Colitis and immune cell infiltration were indistinguishable between control and C/EBPα mutants (Fig S4A and B). The C/EBPα-depleted colitis-associated low-grade dysplasia had high nuclear β-catenin levels, although not significantly, as compared to the control (Fig 5C, quantification Fig S4C). Collectively, AOM-DSS–induced colitis-associated carcinogenesis increases Wnt signaling and reduces C/EBPα expression. C/EBPα depletion further promotes tumor growth in colitis-associated and Wnt signaling-dependent cancer.

### C/EBPα controls proliferation in intestinal organoids

To identify the C/EBPα-regulated genes, we examined the intestinal organoid cultures from C/EBPα$^{Flox/Flox}$-VilinCre$^{ERT2}$ (conditional C/EBPα KO) and C/EBPα$^{Flox/Flox}$ (control) mice. While the control organoids had regular structures after 4-OHT administration, such as extended arms and rounded luminal parts, the homozygous C/EBPα KO organoids grew faster, shown by individual tracked organoids over a period of 4 d and measured by the increase in cell number (Fig 6A and B). EdU labelling of S-phase cells revealed that C/EBPα-depleted organoids had extended proliferative zones in comparison with the controls, where proliferative cells were found exclusively in the crypt-like structures (Fig 6C). RNA was isolated from C/EBPα KO and control organoids and processed for RNA sequencing. Gene set enrichment analysis (GSEA) (Mootha et al, 2003; Subramanian et al, 2005) was performed on the differentially expressed genes in the C/EBPα KO and control organoids. The three top-enriched "hallmark" gene sets included targets for the MYC, E2F and G2M checkpoint genes (Fig 6D). Also, Wnt target genes were significantly enriched by testing for a gene set from APC-mutant mice (Fig 6D). We identified several differentially regulated genes that participate in cell proliferation and that are controlled by Wnt signaling, including cyclin D1 (*Ccnd1*), *Ccne*, *Myc*, *Cdk2*, *Axin2*, *E2f4*, *Macc1*, *Bambi*, and *Cd44*. The data reveal that C/EBPα is involved in regulating genes controlling cell proliferation in intestinal epithelia. Among the down-regulated genes in C/EBPα KO organoids, we found Ptk6 (protein tyrosine kinase 6) that has been shown to negatively regulate Wnt signaling in the gastrointestinal tract by interfering with the interaction between β-catenin and Cdc73 of the Paf1C transcriptional elongation complex (Shi et al, 1997; Palka-Hamblin et al, 2010; Kikuchi et al, 2016). We confirmed a reduction in Ptk6 expression upon loss of C/EBPα by qRT–PCR of independent C/EBPα KO organoids (Fig 6E). Collectively, our data suggest that C/EBPα restricts β-catenin signaling and proliferation in intestinal organoid cultures.

### Caco-2 cells down-regulate C/EBPα after activation of canonical Wnt signaling

We examined C/EBPα expression in human CRC cell lines (LoVo, SW480, LIM1215, HCT116, SW620, HCA7, DLD1, Caco-2). C/EBPα expression was low in five of the eight most commonly used human CRC cell lines. C/EBPα expression was highest in the Caco-2 cells, which are amenable to differentiation in vitro (Fig S5A). Treatment of Caco-2 cells with the GSK3β inhibitor CHIR99021 stabilized β-catenin, as assessed by increased *Axin2* expression and concomitantly reduced *Cebpa* expression (Figs 7A and S5B). As densely grown Caco-2 cells spontaneously differentiate into enterocytes (Pinto et al, 1983; Rousset, 1986; Hidalgo et al, 1989), we monitored C/EBPα expression at different growth states. C/EBPα levels were highest at the onset of differentiation at day 8. C/EBPα expression subsequently declined over a 15-d period (Fig 7B). These findings support the idea that C/EBPα participates in controlling cell differentiation.

HCT116 cells expressed a low level of C/EBPα (Fig S5A). To assess the role of C/EBPα in a Wnt-activated CRC cell line, we generated a stable conditional C/EBPα expression HCT116 cell line that expresses C/EBPα following doxycycline administration (Fig S5C). Activation of the *Cebpa* transgene reduced the clonogenicity of the

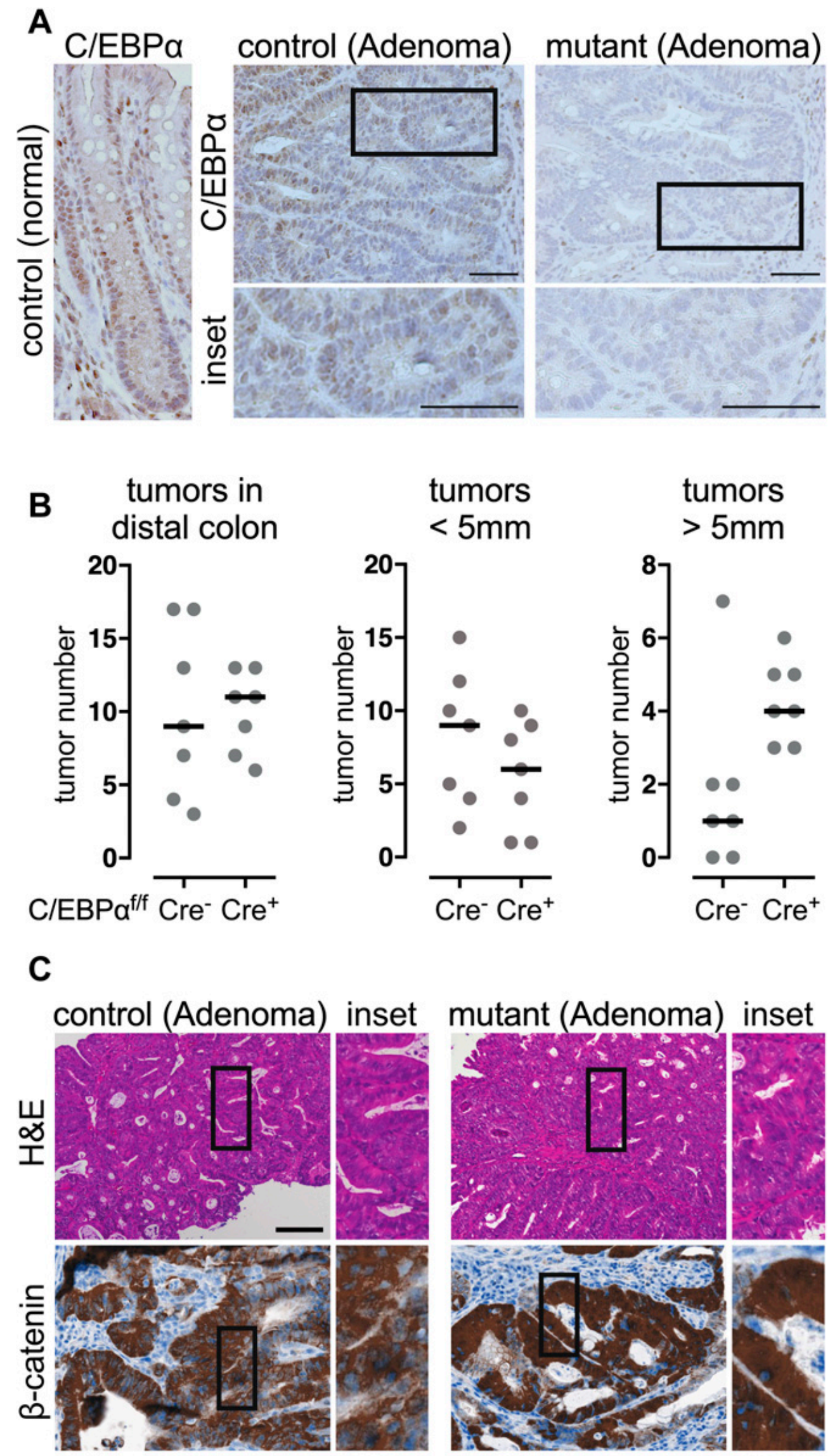


**Figure 5. Loss of C/EBPα promotes tumorigenesis in the AOM/DSS colitis-associated cancer model.**
**(A)** C/EBPα IHC of paraffin sections from control and C/EBPα-depleted (mutant) adenoma; bottom panels show magnified insets. **(B)** Quantification of tumor numbers (left) and tumors >5 mm (right) in control and C/EBPα-depleted colons (in the mutants, one tumor was >10 mm) (two-tailed Mann–Whitney test) n (individual mouse) = 7, $P$ = 0.0245). **(C)** H&E staining (top) and IHC (bottom) of β-catenin in control and C/EBPα-depleted (mutant) adenoma. Scale bars: 100 μm.

HCT116 cells and impaired colony growth (Fig 7C and D). Taken together, our data reveal that C/EBPα and canonical Wnt signaling are opponents in epithelial growth control and suggest a tumor suppressor function of C/EBPα in Wnt-dependent tumorigenesis in the mammalian gut.

## Discussion

CRC is a major burden on health systems worldwide. In recent years, great progress has been made toward elucidating the underlying mechanisms of colon carcinogenesis. Yet, more parts of the puzzle related to signaling, gene regulation, and proliferation control need to be understood in the exploration of novel pharmacological and genetic targets for treating CRC (Vogelstein et al, 2013; Tape, 2017). Here, we show that C/EBPα is expressed in normal gut tissue but is absent in Wnt-activated human CRC cells and murine APC$^{Min/+}$ polyps. Our data support the premise that (i) high C/EBPα and high Wnt expression states are inversely correlated, (ii) C/EBPα reduces oncogene dependent growth, and (iii) C/EBPα plays a tumor-suppressive role in carcinogenesis. Therefore, the data show that C/EBPα has a critical function in CRC pathogenesis and suggests a regulatory Wnt–C/EBPα axis in the gut.

CRC is initiated by gatekeeper mutations such as the Wnt signaling component *APC*. Current hypotheses suggest that cancerous lesions progress from adenoma to carcinoma by acquiring additional sequential mutations over time. This involves genetic alterations that inactivate tumor suppressor genes and activate oncogenes (Fearon, 2011). However, a compilation of tissue-specific suppressors of tumorigenesis is far from complete and C/EBPα may qualify as one of them (Flodby et al, 1996; Schuster & Porse, 2006; Koschmieder et al, 2009; Lourenco & Coffer, 2017). Besides genetic changes, the activity or expression of other non-mutated regulators is altered. We observed reduced C/EBPα expression in an APC$^{Min/+}$ mouse model and in human CRC specimens, where C/EBPα was only detected in cells with absent or low oncogenic β-catenin expression. C/EBPα expression was inversely correlated with cells with tumor propagating potential. Adenomas and adenocarcinomas showed areas of absence of C/EBPα expression in most cases and in particular in the more advanced tumor stages. Low C/EBPα levels have been observed in breast cancer (Gery et al, 2005), and there is epigenetic silencing in acute myelogenous leukemia (Hackanson et al, 2008) which together with our data suggest a general role of C/EBPα as a tumor suppressor gene.

Our histopathological data show that C/EBPα expression and high Wnt/β-catenin signaling are mutually exclusive in intestinal cancer. The experimental and genetic evidence from the mouse gut and organoids contributes mechanistic evidence for the inverse relationship between C/EBPα and activated Wnt signaling, in agreement with the observations of others in adipogenesis and osteoblastogenesis (Kang et al, 2007; Kawai et al, 2007). Our data argue for a feedforward loop of reduced C/EBPα expression in Wnt-dependent tumorigenesis.

Using an AOM-DSS colitis-associated cancer model, we provide further evidence for the relation between tumor size and C/EBPα expression; the C/EBPα–Wnt regulatory axis might be the underlying mechanism. C/EBPα loss primes for high Wnt susceptibility, while Wnt/β-catenin signaling activation with AOM/DSS induces tumorigenesis (Greten et al, 2004). We anticipate that low levels or absence of C/EBPα increase the risk of inflammatory bowel disease or severe inflammation in evolving colitis-associated cancer. Besides the severity of inflammation and genetic alterations, epigenetic factors such as DNA methylation contribute to the development of colitis-associated cancer, as observed by epigenome-wide changes. DNA methyltransferases control gene expression by methylating the cytosine pyrimidine ring in the CpG-rich regions of regulatory genomic units (Ventham et al, 2016; Emmett et al, 2017). In osteogenesis and adipogenesis and in acute

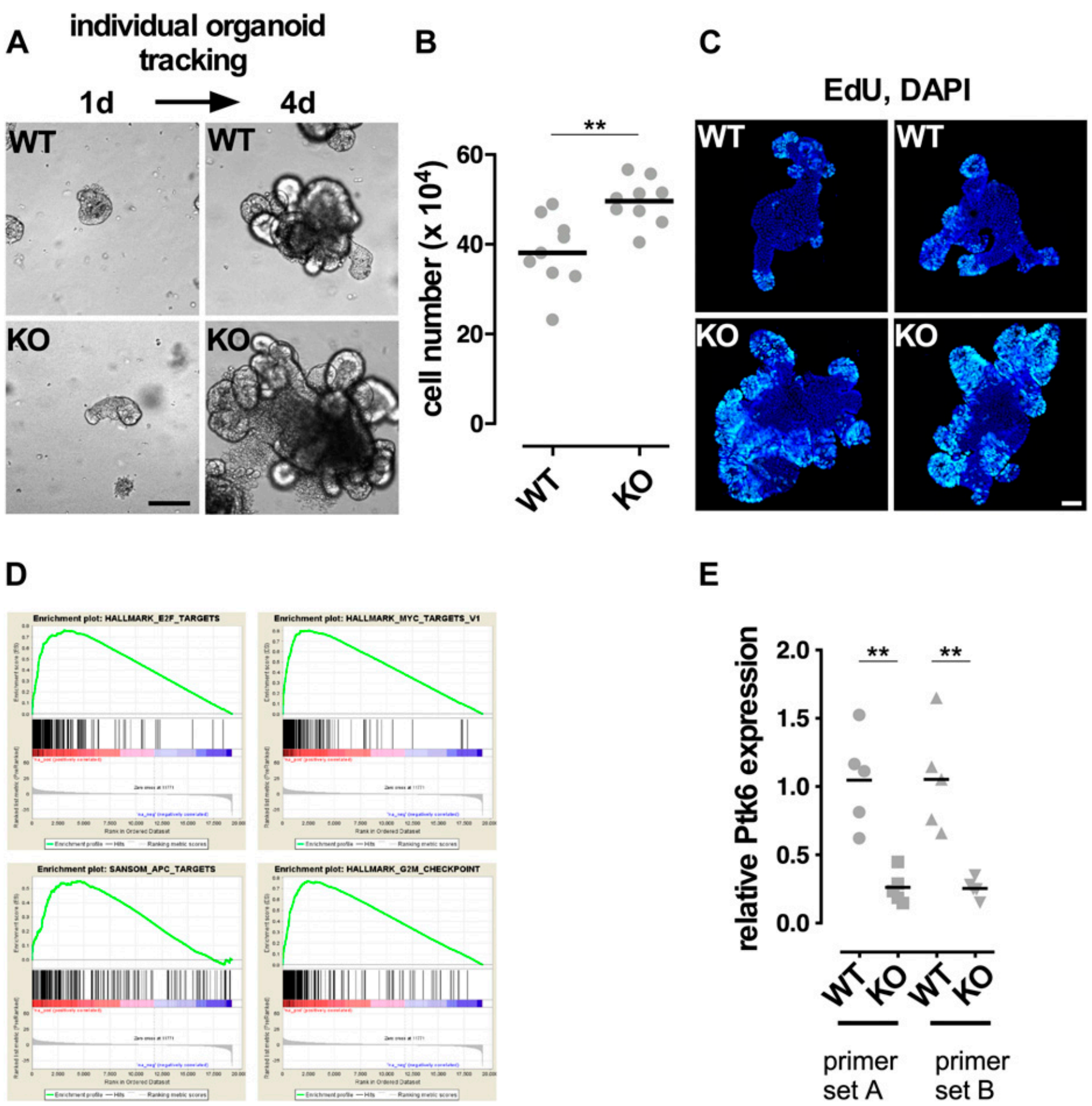


**Figure 6. Analysis of C/EBPα-depleted organoids.**
Comparison of control and C/EBPα-depleted small intestinal organoid cultures. C/EBPα KO was induced over 2 consecutive days by administration of 800 nM 4-OHT. **(A)** Brightfield images of individual tracked control and mutant organoids over a period of 4 d. **(B)** Measurement of total cell number increase in control and mutant Organoids over a period of 4 d (two-tailed, unpaired *t* test, n = 9, $P < 0.05$). **(C)** Whole-mount IF of EdU-labeled S-phase cells of control (upper) and C/EBPα-depleted (lower) small intestinal organoids. **(D)** GSEA of RNA sequencing expression data of control and C/EBPα-depleted small intestinal organoids: C/EBPα depletion results in enhanced expression of the *E2f*, *Myc*, and *Apc* target genes and G2M checkpoint genes. KO: C/EBPα-depleted. **(E)** Quantitative normalized PCR analysis of Ptk6 gene expression in 5 WT and C/EBPα KO organoids with two primer sets, as indicated (two-tailed, unpaired *t* test, n = 5, $P < 0.005$). Scale bars: 200 μm.

myelogenous leukemia, hypermethylation of the CpG islands at the proximal promoter region of *CEBPA* silences C/EBPα transcriptionally (Jost et al, 2009; Gao et al, 2015). A study of DNA methylation differences also reported reduced C/EBPα in patients with colon cancer (Silviera et al, 2012). Together, these data suggest that inflammation initiates epigenetic changes, including DNA methylation, that reduce C/EBPα expression. Reduced C/EBPα expression increases the risk of developing cancer and colitis-associated cancer.

A previous developmental study of intestines from newborn and neonatal C/EBPα-null mice, which die within 8 h after birth by hypoglycemia, revealed no essential role in the morphological maturation of the early developing intestine (Oesterreicher et al, 1998; Wang et al, 2013). However, fetal and adult intestines exhibit strong differences in morphology and gene expression (Crosnier et al, 2006; Nigmatullina et al, 2017). Wnt/β-catenin–dependent stem cells in the intestinal crypt compartment continuously renew the fully developed intestinal epithelium. The progeny proliferate and differentiate in the transient proliferation zone of the crypt and continuously renew the intestinal epithelial barrier (Leblond & Walker, 1956; Potten & Loeffler, 1990; Korinek et al, 1998; Gregorieff et al, 2005; de Lau et al, 2007). C/EBPα expression was very low or absent in the Wnt-dependent intestinal Lgr5 stem cells and Wnt-dependent Paneth cells, but was expressed in the cells of the transient proliferation zone. Therefore, C/EBPα may participate in decreasing the Wnt response, controlling TA zone proliferative expansion, and regulating timely differentiation. This premise is supported by the observation of the Wnt–C/EBPα antagonism in Caco-2 cells that increased Wnt activity reduces C/EBPα expression in the cells, which triggers the Wnt–C/EBPα feedforward loop. We provide genetic evidence that C/EBPα participates in controlling proliferation and the cell cycle regulatory genes. Hyperplasia and adenoma formation occur also via the loss of APC in cells with normally reduced transcriptional Wnt response (Powell et al, 2012; Metcalfe et al, 2014). Therefore, preceding low C/EBPα expression may promote Wnt-dependent cancer initiation, proliferation, and tumor progression. Based on organoid cultures, our data support a mechanism, in which C/EBPα participates in the regulation of Wnt/β-catenin signaling by controlling expression of Ptk6. Ptk6 is expressed in intestinal crpyts and promotes apoptosis by inhibiting prosurvival signaling in response to DNA damage (Haegebarth et al, 2009). Ptk6 phosphorylates Cdc73 (parafibromin, a component of the RNA polymerase II–associated Paf1C complex) to negatively regulate β-catenin/TCF transcription (Shi et al, 1997; Palka-Hamblin et al, 2010; Kikuchi et al, 2016). Ptk6 expression is reduced in human adenocarcinoma, and reduction in Ptk6 also promotes the growth of xenografts (Mathur et al, 2016). In conclusion, C/EBPα might attenuate Wnt/β-catenin signaling and impact on cancer cell proliferation by controlling expression of Ptk6.

Tight control of Wnt responsiveness is critical for regulating crypt compartment proliferation and differentiation. The distance to Wnt

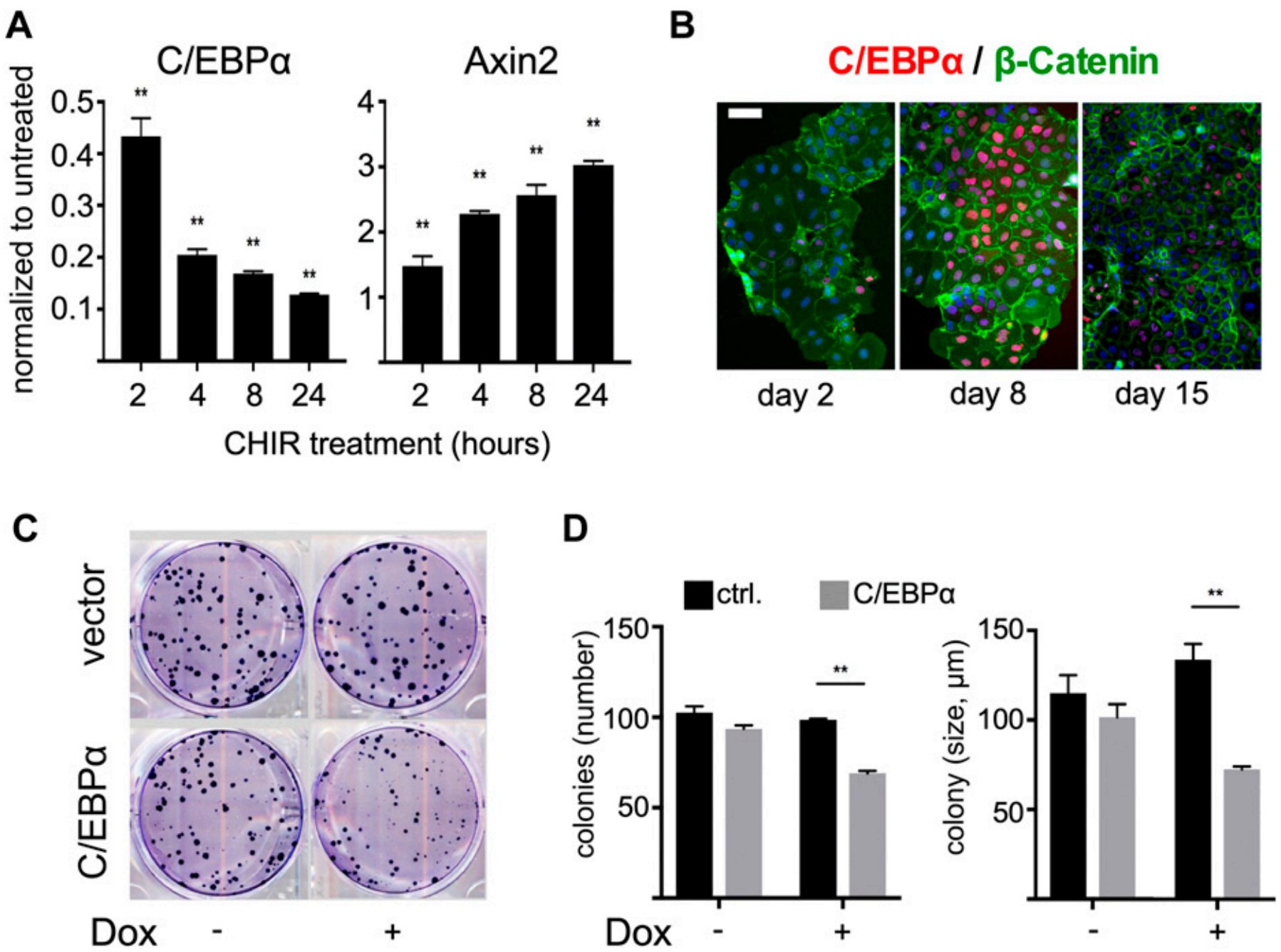


**Figure 7. C/EBPα in CRC cell lines.**
**(A)** mRNA expression of *CEBPA* and the Wnt target *AXIN2* in Caco-2 cells upon activation of Wnt signaling by inhibition of GSK3β. There is an inverse correlation between C/EBPα expression and Wnt signaling activity (AXIN2). **(B)** C/EBPα (red) expression levels at different Caco-2 cell differentiation steps. **(C)** Colony formation assay of control and doxycycline (300 ng/ml)-induced C/EBPα-expressing HCT116 cells. **(D)** Quantification of colony number and size showing that C/EBPα expression reduces clonogenicity and colony growth (** $P < 0.01$).

ligand–producing cells in the lower part of the crypt and active BMP signaling prevent Wnt activation in epithelial cells of the villi (Crosnier et al, 2006). Both Wnt signaling and C/EBPα expression are low in differentiated epithelial cells of the villi. C/EBPα appears to be dispensable in fully differentiated cells, where Wnt does not control its expression. Loss of C/EBPα expression in the villus cells is likely to occur by epigenetic mechanisms, potentially by DNA methylation, as observed in leukemic cells, which also demonstrates the central importance of C/EBPα expression in other neoplasms (Bennett et al, 2007; Hackanson et al, 2008; Jost et al, 2009; Lu et al, 2010; Lin et al, 2011; Di Ruscio et al, 2013; Gao et al, 2015). In conclusion, we show that the loss of C/EBPα expression is a crucial step in the initiation and growth of colorectal neoplasms and is in line with the findings in other tumor entities.

# Materials and Methods

### APC$^{Min/+}$ mice and tissue preparation

C57BL/6J-Apc$^{Min}$/J mice were purchased from Jackson Laboratories. C/EBPa floxed mice originate from Claus Nerlov, Ex3-β-catenin floxed from Makoto M Taketo (Harada et al, 1999), VillinCreERT2 from Sylvi Robine (el Marjou et al, 2004), and Lgr5-GFP reporter mice from Hans Clevers (Barker et al, 2007). All mice were housed in individually ventilated cages in a specific pathogen–free mouse facility at the Max Delbrück Center for Molecular Medicine, Berlin. The local government authority (Landesamt für Gesundheit und Soziales Berlin [LaGeSo], Germany) approved the animal studies. Colitis-associated tumorigenesis and depletion of C/EBPα was induced by intraperitoneal injection of tamoxifen (50 mg/kg body weight) on 5 consecutive days, 7 d later by 1× intraperitoneal injection of 12,5 mg/kg azoxymethane, and 3 intervals of 1 wk of 2% DSS in drinking water. The mice were euthanized by cervical dislocation at protocol defined time points or when they showed signs of disease, and the organs were quickly dissected, flushed with cold PBS, and fixed overnight in 4% formalin for paraffin embedding, or stored in RNAlater (Ambion) for RNA extraction. To assess macroscopic tumors in the intestine (>0.5 mm), the intestinal tract was removed immediately after euthanasia, divided into four segments comprising the duodenum, jejunum, ileum, and colon, opened longitudinally, rinsed with cold PBS, and examined under a dissection microscope.

### Intestinal organoid culture, fixation, and paraffin embedding

Intestinal organoid culture was performed as described previously (Sato et al, 2009; Heuberger et al, 2014). Briefly, jejunal crypts were isolated by filtration (70 μm) and centrifugation (400 *g*/3 min) of selected fractions after mechanical dissociation (shaking) of the villi and crypts after 5-min incubation at room temperature with 8 mM and 2 mM EDTA and at 25-min rotation at 4°C, respectively. We embedded 400 crypts in 50 μl Matrigel (BD, 356231) and cultured them in DMEM/F12 medium (12634; Life Technologies) supplemented with N2 and B27 (17502-040 and 17504-044, respectively; Life Technologies), mNoggin (Cat. No. 250-38, final concentration 100 ng/ml; PeproTech), mEGF (mouse epidermal growth factor, PMG 8041, final concentration 50 ng/ml; Life Technologies), hrSpo1 (human rSpo1, Cat. No. 120-38, final concentration 100 ng/ml; PeproTech), and acetylcysteine (A9165, final concentration 1.25 mM; Sigma-Aldrich). Cre-mediated recombination was induced by administering 800 nM 4-OHT for 2 consecutive days.

Growth of individual organoids was tracked with a Leica DIM6000 microscope equipped with an NPlan 10× NA 0.25 objective and a

motorized LMT200 V3 High precision Scanning Stage to relocate multiple times previously stored positions. Growth of organoids by total cell numbers was measured with the NucleoCounter NC-200 from Chemotec. Defined cell numbers of organoids were seeded and cultured for 4 d. Cells of organoids were harvested directly from the Matrigel using buffer A100 (4 min incubation and repeated trituration) and buffer B according to the manufacturer's description.

#### *Fixation and sectioning*

Organoids containing Matrigel were disintegrated by trituration and transferred to 5 ml of cold DMEM/F12 medium. After centrifugation, the organoids were resuspended for 3 h in 4% PFA/PBS. The fixative was exchanged with PBS, and the organoids were embedded in 2% agarose/PBS and transferred to 70% ethanol, followed by paraffin embedding. Paraffin sections (5–10 $\mu$m) were obtained for histological analysis.

### RNA extraction, cDNA, and real-time qRT–PCR

Total RNA was isolated from cells and tissues using GeneMATRIX Universal RNA Purification Kit (Roboklon) according to the manufacturer's instructions. A DNase I digest was included. RNA concentrations were quantified with a NanoDrop spectrophotometer (Thermo Fisher Scientific). Total RNA (1 $\mu$g) was reverse-transcribed with oligo(dT) primers using SuperScript II enzyme (Thermo Fisher Scientific) according to the manufacturer's instructions. PCR was performed using a primer/probe-based TaqMan system with the housekeeper run in duplex in the same well. Standard protocols and settings were used. The primer/probe mixes used were for murine *Cebpa* Mm00514283_s1 and murine $\beta$-actin (*Actb*). Relative mRNA expression values were calculated using the $\Delta\Delta C_t$ (comparative threshold cycle) method.

### RNA sequencing and GSEA

RNA was isolated and processed for RNA sequencing. RNA quality control was performed using BioAnalyzer (Agilent). Sequencing libraries were prepared using a TruSeq Stranded mRNA kit (Illumina). Paired-end sequencing (2 × 75 nt) was performed using an Illumina HiSeq 4000 system (TruSeq PE Cluster kit, TruSeq 300 cycle kit). We obtained 32.9–40.8 M (37.1 ± 2.6) sequencing reads per sample. Read quality was controlled using FastQC software (Andrews, 2010) followed by Bowtie 2 (v. 2.2.9)-based mapping (Langmead & Salzberg, 2012) and RSEM (v. 1.2.31)-based quantification (Li & Dewey, 2011). Differential expression analysis was performed using the DESeq2 (v.1.14.1) package in R (Love et al, 2014).

The Molecular Signature Database MSigBD (Liberzon et al, 2011) metagene sets "hallmark" and "curated (C2)" were used to apply the camera tool (Wu & Smyth, 2012) on the voom-transformed (Law et al, 2014) count data using a limma-based (Smyth, 2005) ranking metric. Gene sets with an adjusted *P*-value < 0.05 were considered significant. The full results are displayed in Supplemental Materials.

### CRC and colorectal adenoma tissues

We obtained tissue sections from subjects with spontaneous intestinal adenoma (n = 8) and/or CRC (n = 11) plus matched (same patient) and non-matched normal mucosa (n = 18). The study received a positive ethics vote from the Friedrich-Alexander- Universität Erlangen-Nürnberg Ethics Commission. TableS1 shows the clinicopathological data.

### IHC and IF

C/EBP$\alpha$ IHC and IF were performed on 5-$\mu$m formalin-fixed, paraffin-embedded tissue sections. All incubation steps were performed at room temperature unless stated otherwise. The sections were deparaffinized (2 × 10 min in Histo-Clear II, National Diagnostics) and hydrated in a descending ethanol series (2 min each in 2 × 100%, 85%, 70%, 50%, and 30% ethanol in double-distilled water [dd$H_2O$], dd$H_2O$). Antigen retrieval was performed by 15-min incubation in pre-heated citrate buffer (pH 6.0) in a microwave, with boiling intervals. Sections were cooled to room temperature for 20 min and washed in PBS-T (Tween 20, 0.02%). If HRP-based detection was performed later, endogenous peroxidases were blocked by 10-min incubation with 5% $H_2O_2$ in methanol. After washing (PBS-T, 2 × 5 min), the sections were incubated with 10% normal serum (from the animal used to generate the subsequently used secondary antibody/ies) in PBS-T. For C/EBP$\alpha$ brightfield IHC, endogenous avidin/biotin blocking was performed as described in the kit manufacturer's manual (Abcam). All antibodies (C/EBP$\alpha$, D56F10, Cell Signaling Technology, 1:100; mucin 2 [MUC2], H-300, Santa Cruz, 1:100; $\beta$-catenin, Clone14, BD Transduction Laboratories, 1:200; Ki67 for mouse clone TEC-3, for human clone MIB1, both Dako; GFP, Abcam, 1:400) were incubated at 4°C overnight in SignalStain antibody diluent (Cell Signaling Technology). After washing (3 × 5 min, PBS-T), the specimens were incubated for 1 h with Alexa488- or Alexa594–coupled anti-mouse, anti-rat, and/or anti-rabbit secondary antibodies (1:1,500; Invitrogen) for IF, or with a biotin-coupled anti-rabbit secondary antibody (111-065-003, 1:500; Jackson Laboratories) for C/EBP$\alpha$ brightfield IHC. After washing (3 × 5 min, PBS-T), IF sections were counterstained with DAPI, washed again (3 × 5 min, PBS-T), and mounted in fluorescent mounting medium (Dako). For brightfield C/EBP$\alpha$ staining, a HRP–streptavidin complex (dianova) was incubated at 2 $\mu$g/ml in PBS-T for 30 min, followed by another round of washing. The protein was visualized by 3–5-min treatment with FAST DAB (Sigma-Aldrich); the reaction was stopped in dd$H_2O$, followed by counterstaining with Mayer's hematoxylin (Carl Roth). After rinsing with tap water and transfer through an ascending ethanol series and Histo-Clear II treatment (2 × 5 min), the sections were mounted using Omnimount (National Diagnostics).

### Cell culture

Caco2 and Hct116 cell were cultured in DMEM, 5% serum, Penstrep (Life Technologies).

Caco-2 cells were treated with 3 nM CHIR99021 (Tocris) for the indicated time. Hct116 cells were stably transduced with pInducer21-C/EBP$\alpha$ lentiviral particles, fluorescence-activated cell sorted for GFP$^{high}$ cells, and C/EBP$\alpha$ expression was induced by doxycycline. pInducer21-C/EBP$\alpha$ was constructed by LR-clonase reaction (Invitrogen Life Technologies) with pENTR2B-hC/EBP$\alpha$ and pInducer21

(Meerbrey, 2011). Lentiviral particles were produced in 293TN cells co-transfected with psPAX2, pMD2.G, and pInducer21-C/EBPα.

## Dataset Availability

Gene expression data supporting the conclusions of this research article are available under GEO accession number GSE123925.

## Supplementary Information


## Acknowledgements

We thank Claus Nerlov, University of Oxford, Medical Research Council, for the floxed C/EBPα mouse strain. We thank Amir Orian for critical comments and as part of the international exchange program "SignGene" of the MDC (Max Delbrück Center for Molecular Medicine). We thank Juliette Bergemann and Daniel Stepczynski from the MDC animal facility, and Frauke Kosel for technical assistance (MDC). The work was supported by funds of the "Berliner Krebsgesellschaft e.V." (HIFF201513).

### Author Contributions

J Heuberger: conceptualization, resources, data curation, formal analysis, supervision, validation, investigation, visualization, methodology, and writing—original draft, review, and editing.
U Hill: conceptualization, data curation, investigation, visualization, methodology, and writing—original draft.
S Förster: data curation, investigation, visualization, and writing—review and editing.
K Zimmermann: data curation, software, and validation.
V Malchin: investigation and methodology.
AA Kuhl: resources, data curation, formal analysis, and visualization.
U Stein: resources.
M Vieth: resources, data curation, formal analysis, investigation, methodology, and writing—review and editing.
W Birchmeier: conceptualization and writing—review and editing.
A Leutz: conceptualization, resources, data curation, formal analysis, supervision, funding acquisition, validation, investigation, visualization, methodology, project administration, and writing—original draft, review, and editing.

### Conflict of Interest Statement

The authors declare that they have no conflict of interest.

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
