## [Reviewer comments · Life Science Alliance]

Life Science Alliance

A C/EBP α -Wnt connection in gut homeostasis and carcinogenesis

Julian Heuberger, Undine Hill, Susann Foerster, Karin Zimmermann, Victoria Malchin, Anja Kuhl, Ulrike Stein, Michael Vieth, Walter Birchmeier, and Achim Leutz

DOI: [10.26508/lsa.201800173](https://doi.org/10.26508/lsa.201800173)

Corresponding author(s): Achim Leutz, Max Delbrück Center for Molecular Medicine

Review Timeline:

Submission Date:	2018-08-24
Editorial Decision:	2018-09-24
Revision Received:	2018-11-29
Editorial Decision:	2018-12-10
Revision Received:	2018-12-17
Accepted:	2018-12-18

Scientific Editor: Andrea Leibfried

Transaction Report:

September 24, 2018

Re: Life Science Alliance manuscript #LSA-2018-00173-T

Prof. Achim Leutz
Max Delbrück Center for Molecular Medicine
Tumorigenesis and Cell Differentiation
Robert-Rössle-Strasse 10
Berlin, Berlin 13125
Germany

Dear Dr. Leutz,

Thank you for submitting your manuscript entitled "A C/EBP α -Wnt connection in gut homeostasis and carcinogenesis" to Life Science Alliance. The manuscript was assessed by expert reviewers, whose comments are appended to this letter.

As you will see, the reviewers appreciate your data but think that your manuscript can be further strengthened by following their suggestions. Importantly, reviewer #1 notes that a control is missing (effects of CEBPA ablation on DSS-induced colitis), and reviewer #2 confirmed during cross-commenting that it'd be good to add this control. The other experiments requested seem to be all straightforward to address, and we would thus like to invite you to provide a revised version of your manuscript as well as a point-by-point response to all concerns raised. Please note that point 10 of ref#2 can be addressed in the discussion only, adding experimental data to respond to this concern is not needed for publication in Life Science Alliance.

Thank you for this interesting contribution to Life Science Alliance. We are looking forward to receiving your revised manuscript.

Sincerely,

- A letter addressing the reviewers' comments point by point.
- An editable version of the final text (.DOC or .DOCX) is needed for copyediting (no PDFs).
- High-resolution figure, supplementary figure and video files uploaded as individual files: See our detailed guidelines for preparing your production-ready images, <http://life-science-alliance.org/authorguide>
- Summary blurb (enter in submission system): A short text summarizing in a single sentence the study (max. 200 characters including spaces). This text is used in conjunction with the titles of papers, hence should be informative and complementary to the title and running title. It should describe the context and significance of the findings for a general readership; it should be written in the present tense and refer to the work in the third person. Author names should not be mentioned.

B. MANUSCRIPT ORGANIZATION AND FORMATTING:

Full guidelines are available on our Instructions for Authors page, <http://life-science-alliance.org/authorguide>

Reviewer #1 (Comments to the Authors (Required)):

A functional antagonism between C/EBPa and Wnt/b-catenin signaling has been described in normal differentiating cells. Moreover there is evidence that C/EBPa acts as a tumor suppressor in various tumor types. The present study appears to be the first one to demonstrate the interplay of

C/EBP α and Wnt/b-catenin signaling in the intestine and colorectal cancer. Since activation Wnt signaling is considered the main initial event in CRC development this study is of high interest to understand colorectal tumorigenesis. While most of the results appear clear-cut, there are a number of shortcomings mostly with respect to style and clarity of presentation and interpretations that need to be amended.

Specific concerns:

1. Figure 1A shows IHC of C/EBP α in human intestinal tissues and tumors. It is stated that C/EBP α is absent from the normal crypt compartment but present in the transit amplifying zone. It is not noted, although evident from the picture, that C/EBP α is also present in the layers above the transit amplifying zone which is in contrast to staining of mouse intestine in Fig. 2 where C/EBP α is restricted to the TAZ. Please clarify.

2. Figure 1B-K shows C/EBP α in human colorectal carcinomas. There are a number of unclear aspects that need to be addressed.

(a) Table 1 does not show expression intensities of the various samples in contrast what is claimed on p. 5, first line.

(b) The description of the panels is confusing. All that one can extract from this is that the expression of C/EBP α in tumors is highly variable. An inverse correlation between C/EBP α levels and tumor stage as claimed in the discussion (p. 10) cannot be drawn from these data. Moreover, it is not getting clear why three panels are shown to illustrate each percentage of expression. Also it appears that expressions in panels D and G are lower than the expressions shown in B,C, and E,F, respectively. What do the numbers written within panels represent? What is the meaning of the dotted line in panel K?

This figure and table 1 need to be revised.

3. Results p. 6 : "However, C/EBP α expression was detected in cells at the transition zone into the adenomatous tissue, which had lower levels of nuclear β -catenin (Figure3, inset, left of dotted line)." What does the term "transition zone" mean in this context? Aren't these just normal epithelial cells? Why not say so?

4. Figure 5A: It is claimed that "...the control adenomatous lesions likewise had reduced C/EBP α expression", but this cannot be concluded from the pictures because normal tissue is not shown. Figure 5B: What is meant by "distal tumors"? Does each point represent tumor number of one individual mouse? If so, please mention in figure legend.

The experimental procedures of the DSS/AOM model are not included in Materials and Methods.

5. Figure 5 shows results of a key experiment revealing that ablation of C/EBP α in the AOM/DSS model yields bigger tumors. However, since tumorigenesis in this model is dependent on the induction of colitis by DSS, altered tumorigenesis could be based on altered colitis after loss of C/EBP and not on direct changes in Wnt/b-catenin signaling. It should be analyzed whether the degree of colitis in DSS treated animals is altered by C/EBP α depletion.

6. Discussion: (a) Where is it shown that C/EBP α "restricts TA zone expansion" (p.10) (b) As mentioned above, a correlation between C/EBP α expression and tumors stage is not evident from the results,

7. There is one more general point to consider: There is good evidence that crypt stem cells are the source of intestinal tumor cells at least in APC-dependent tumorigenesis. If this is true for the AOM/DSS model as well then it would not be astonishing that C/EBP α is absent from tumors because it is already absent or low in the stem cells as shown by the authors. More importantly, in that case any effects of C/EBP ablation as shown in Figure 5 would be indirect, e.g. due to loss of C/EBP α in other cell types such as the ones in the transit amplifying zone. Alternatively, C/EBP is only reduced but not absent from crypt cells. Indeed, results from organoids (derived from crypts) where loss of C/EBP has pronounced effects on cell proliferation and Wnt target gene expression suggest that this is the case. The authors should determine the expression status of C/EBP in wild-type and mutant crypts using more sensitive methods than IHC, such as RT-PCR, and discuss their

results accordingly.

Minor:

The term "GSK3b destruction complex" is rather uncommon in the field (p. 5). Replace by b-catenin destruction complex.

Reviewer #2 (Comments to the Authors (Required)):

Heuberger et al describe an inverse correlation between C/EBPa expression and the level of WNT/b-catenin signaling in intestinal tumors, organoid cultures, and CRC cell lines. In general, the paper is well written and easy to follow, the data are well presented and, with a few exceptions (noted below), the claims made in the paper support the conclusions drawn. The interaction between WNT and C/EBP has been established in other systems, but this work is a nice demonstration of how it may impact intestinal biology and tumor development.

Specific points:

1. Human tumor and normal samples were graded for C/EBPa expression, but the actual data (expression grades) are not presented anywhere I could find - only the comparison statistic in Table I. I think it is appropriate to show the scoring - however the authors deem it best to convey the raw data.
2. The authors make the statement: "cells with the most EdU labeling had the lowest C/EBPa expression" to conclude an "inverse relationship between S-phase and C/EBPa expression". However, EdU/BrdU incorporation is more of a binary process - if cells are uptaking the analog they are going through S-phase, less signal may mean they are progressing more slowly, or later in the process, but still in S-phase. Can the author clarify this statement? Perhaps it was meant to imply that C/EBPa positive cells less frequently show EdU uptake?
3. On Page 7, the authors state " [organoids] had reduced villi-like areas, and formed larger cells and crypt buds". I can't find any direct data to support this, other than one example image in Figure 4. To make this claim would require further examples and quantitation of crypt bud number, cells size, and differentiated area.
4. The AOM experiment shows some data to support the notion that C/EBPa KO's develop larger tumors, although the data is represented as tumors >5mm. Were the tumors actually measured and can the authors provide the continuous data? Also - please include the AOM/DSS details in the methods section.
5. Page 8: "The C/EBPa-depleted colitis associated low grade dysplasia had high nuclear b-catenin levels, although not significantly, as compared to control (Figure 3C)". do the authors mean Figure 5C? And I also cannot find any quantitation of b-catenin levels in this experiment to provide data for this conclusion.
6. Also page 8: The authors refer to "APC target genes". APC is not a transcription factor - I assume this means WNT target genes, but it is important to define this properly.

7. In the list of CRC cell lines, I have never heard of DCD1. Do the authors mean DLD1?

8. Page 9: "Activation of the Cebpa transgene abrogated the clonogenicity of the HCT116...". I disagree with the use of the word 'abrogate'. This implies the effect was profound and almost complete, but in reality I would say clonogenicity was moderately reduced. It is quantified, so stating the absolute effect would be appropriate.

9. Figure 6: The authors claim that the C/EBPa KO organoids had "increase crypt bud number and size" but provide no quantitation on this. Absolutely required to make this statement (and from multiple independent organoid isolates - they can vary line to line).

10. Finally, there is one sentence in the discussion/summary that raises multiple concerns for how the data was interpreted. It is: "Our data support the premise that C/EBPa: i) controls proliferation and regulated WNT target genes, ii) plays a tumor suppressive role in carcinogenesis, and iii) restricts TA zone expansion in the gut."

First, I don't think any direct evidence is presented that C/EBPa regulates WNT target genes. What is present is an inverse correlation between C/EBPa and a high WNT state. Much more experimental data would be required to say it is directly regulating WNT target genes - one easy experiment would be to use the inducible system shown in Figure 7 and determine whether acute C/EBPa induction alters WNT target genes. A further step would be to show direct binding by ChIP.

Second, "restrict TA zone expansion in the gut". Is this true outside of organoids (which were not quantified)? If so, I would imagine that the Villin-Cre/CEBPaFL/FL mice would show and expanded TA zone in the normal intestine/colon. Do the authors have any data to support this?

I don't think these issues require extensive experimental data, but I don't think the current language is supported by the data.

Point-by-point response

We would like to thank the reviewers for their critical insightful comments and suggestions that have helped to improve and further strengthened our manuscript.

Reviewer #1 (Comments to the Authors (Required)):

A functional antagonism between C/EBP α and Wnt/b-catenin signaling has been described in normal differentiating cells. Moreover there is evidence that C/EBP α acts as a tumor suppressor in various tumor types. The present study appears to be the first one to demonstrate the interplay of C/EBP α and Wnt/b-catenin signaling in the intestine and colorectal cancer. Since activation Wnt signaling is considered the main initial event in CRC development this study is of high interest to understand colorectal tumorigenesis. While most of the results appear clear-cut, there are a number of shortcomings mostly with respect to style and clarity of presentation and interpretations that need to be amended.

Point-by-point response to reviewer #2:

Specific concerns:

1. Figure 1A shows IHC of C/EBP α in human intestinal tissues and tumors. It is stated that C/EBP α is absent from the normal crypt compartment but present in the transit amplifying zone. It is not noted, although evident from the picture, that C/EBP α is also present in the layers above the transit amplifying zone which is in contrast to staining of mouse intestine in Fig. 2 where C/EBP α is restricted to the TAZ. Please clarify.

Indeed, in the human colonic crypt C/EBP α is also expressed in the upper part of the crypt. We corrected and clarified this issue in the manuscript. See also Point 7, reviewer #1.

“..... In the normal human colon epithelium, C/EBP α was expressed in the nuclei in the transient proliferation zone and differentiated cells but was largely absent from cells at the base of the crypts (Figure 1A).” (p.4)

2. Figure 1B-K shows C/EBP α in human colorectal carcinomas. There are a number of unclear aspects that need to be addressed.

(a) Table 1 does not show expression intensities of the various samples in contrast what is claimed on p. 5, first line.

(b) The description of the panels is confusing. All that one can extract from this is that the expression of C/EBP α in tumors is highly variable. An inverse correlation between C/EBP α levels and tumor stage as claimed in the discussion (p. 10) cannot be drawn from these data. Moreover, it is not getting clear why three panels are shown to illustrate each percentage of expression. Also it appears that expressions in panels D and G are lower than the expressions shown in B,C, and E,F, respectively. What do the numbers written within panels represent? What

*is the meaning of the dotted line in panel K?
This figure and table 1 need to be revised.*

We thank the reviewer for pointing out that a central figure in our manuscript requires amendment. We have re-organized Figure 1 and provided a new Supplemental Table 1 that lists all patient specific data and details of C/EBP α expressing areas and the scores of the intensity of C/EBP α expression. Indeed, the expression levels of areas of C/EBP α expression in tumors is variable, however, the area of C/EBP α immune reaction and its overall level clearly decreases in tumor tissue in comparison to normal, healthy tissue. Furthermore, tumor bearing tissue often is a mix of tumor cells / adjacent to normal tissue. To make these facts more transparent, we included a new Supplemental Table 1. Further, we removed redundant microphotographs and incorporated data previously shown in Table 1 as a graph (Figure 1F). Data in the graph are shown as individual data points and the Mean with p-scores above. A paragraph describing Figure 1 was adjusted accordingly (page 4-5). The findings need to be considered together with the inverse correlation of C/EBP α / β -catenin expression level in cells and in particular with the observation that cells with oncogenic potential that express β -catenin in the nucleus were never seen to express C/EBP α . According to the reviewer's suggestion, we confined the claim of the inverse correlation between β -catenin and C/EBP α expression:

".....and in human CRC specimens, where C/EBP α was only detected in cells with absent or low oncogenic β -catenin expression. C/EBP α expression was inversely correlated to cells with tumor propagating potential. ..." (p.11)

(reviewer #1, Point 2b, end of paragraph): The small numbers written within the panels represented the individual sample/patient numbers, however, Figure 1 has been adjusted and the issue is therefore resolved. The dotted line in panel K (now Figure 1E) represents the border of cancerous tissue and adjacent neighboring healthy tissue (now been mentioned in the legend to the figure). (p. 25)

3. Results p. 6 : "However, C/EBP α expression was detected in cells at the transition zone into the adenomatous tissue, which had lower levels of nuclear β -catenin (Figure3, inset, left of dotted line)." What does the term "transition zone" mean in this context? Aren't these just normal epithelial cells? Why not say so?

We aimed to describe the border of adenoma and normal tissue that was clearly distinguishable by β -catenin and C/EBP α staining. We rephrased this sentence.

".... However, adjacent to the adenomatous tissue C/EBP α expression was detected in normal cells that had lower levels of nuclear β -catenin..." (p. 6)

4. *Figure 5A: It is claimed that "...the control adenomatous lesions likewise had reduced C/EBP α expression", but this cannot be concluded from the pictures because normal tissue is not shown.*

We included an image showing IHC staining for C/EBP α of adjacent, healthy and not recombined tissue of the same section.

*Figure 5B: What is meant by "distal tumors"? Does each point represent tumor number of one individual mouse? If so, please mention in figure legend.
The experimental procedures of the DSS/AOM model are not included in Materials and Methods.
What is meant by "distal tumors"?*

What is meant by "distal tumors"?

The term "distal tumors" was used for tumors at the distal part of the colon.

Does each point represent tumor number of one individual mouse?

Yes, one point represents the tumor number of one individual mouse. We added this information to the figure legend (p. 26).

The experimental procedures of the DSS/AOM model are not included in Materials and Methods.

We thank the reviewer for pointing this out and we apologize for the neglect. The description of the DSS/AOM model has now been added to the manuscript. (p. 14)

5. *Figure 5 shows results of a key experiment revealing that ablation of C/EBP α in the AOM/DSS model yields bigger tumors. However, since tumorigenesis in this model is dependent on the induction of colitis by DSS, altered tumorigenesis could be based on altered colitis after loss of C/EBP and not on direct changes in Wnt/b-catenin signaling. It should be analyzed whether the degree of colitis in DSS treated animals is altered by C/EBP α depletion.*

We performed colitis scoring and staining for immune cells. Quantification of staining for immune cells: B cells (B220), IgA, monocytes and macrophages (F4/80), T-cells (Cd4 and Cd8); revealed no significant differences between mutant and controls. These data indicate but do not entirely rule out that in C/EBP α mutants increased tumor growth is not triggered by increased infiltration of immune cells. We included these data in our manuscript as Supplemental Figure 4 B, C (page 8):

"..Colitis and immune cell infiltration was indistinguishable between control and C/EBP α mutants (Figure S4 B, C)." (p. 8)

6. Discussion: (a) Where is it shown that C/EBP α "restricts TA zone expansion" (p.10) (b) As mentioned above, a correlation between C/EBP α expression and tumors stage is not evident from the results.

- a) The "restriction to the TA zone" was deduced from i) C/EBP α expression in the transit amplifying cells of mouse small intestine and ii) results of the Edu labeling experiment in organoids in combination with RNA-seq data showing differential regulation of cell cycle genes. However, to clarify this issue we rephrased the sentence to:

"...,ii) C/EBP α reduces oncogene dependent growth"

- b) We already addressed this issue in point 2.
We agree with the reviewer that stating a correlation with tumor stage may be a matter of interpretation. We therefore re-phrased this aspect in the discussion:

"....and in human CRC specimens, where C/EBP α was only detected in cells with absent or low oncogenic β -catenin expression. C/EBP α expression was inversely correlated to cells with tumor propagating potential. ..." (p. 11)

7. There is one more general point to consider: There is good evidence that crypt stem cells are the source of intestinal tumor cells at least in APC-dependent tumorigenesis. If this is true for the AOM/DSS model as well then it would not be astonishing that C/EBP α is absent from tumors because it is already absent or low in the stem cells as shown by the authors. More importantly, in that case any effects of C/EBP ablation as shown in Figure 5 would be indirect, e.g. due to loss of C/EBP α in other cell types such as the ones in the transit amplifying zone. Alternatively, C/EBP is only reduced but not absent from crypt cells. Indeed, results from organoids (derived from crypts) where loss of C/EBP has pronounced effects on cell proliferation and Wnt target gene expression suggest that this is the case. The authors should determine the expression status of C/EBP in wild-type and mutant crypts using more sensitive methods than IHC, such as RT-PCR, and discuss their results accordingly.

We appreciate the critical comments and for bringing up the aspect of intestinal stem cells as the source of intestinal tumor cells. We refined our immunofluorescence staining for C/EBP α in paraffin sections of Lgr5-GFP reporter mice and imaged using confocal spinning disc microscopy to detect also very weak expression. The results were quantified using Imaris software and spot detection. The additional data (now shown in Supplemental Figure 2) revealed that C/EBP α expression is indeed very low at the crypt base and increases towards the top of the crypt. We detected low expression of C/EBP α in Lgr5-stem cells and very weak to none in Paneth cells. Thus, the data may argue against a presumptive indirect effect of LOF C/EBP α in tumorigenesis. We adjusted Supplemental Figure 2 by adding a graphical description of the quantification and a diagram summarizing the measurement (Supplemental Figure 2).

Further, we added staining for C/EBP α of a non-diseased area. Showing that in control colitis associated dysplasia C/EBP α is only reduced. Accordingly, we changed the manuscript on page 5:

"...C/EBP α was weakly expressed in Lgr5-positive stem cells at the bottom of the crypts (Figure 2A, arrow heads), but was absent from lysozyme-positive Paneth cells and terminally differentiated cells of the villus. Expression levels in the crypt were quantified from immunofluorescence images comparing Lgr5-stem cells and other cells crypt cells of the region 1 to +5 cell, 6 to +8 cell and 9 to +12 cells (Figure S2)."... (p. 5)

Minor:

The term "GSK3b destruction complex" is rather uncommon in the field (p. 5). Replace by b-catenin destruction complex.

The terminology was altered according to the suggestion of reviewer #1.

Reviewer #2 (Comments to the Authors (Required)):

Heuberger et al describe an inverse correlation between C/EBP α expression and the level of WNT/b-catenin signaling in intestinal tumors, organoid cultures, and CRC cell lines. In general, the paper is well written and easy to follow, the data are well presented and, with a few exceptions (noted below), the claims made in the paper support the conclusions drawn. The interaction between WNT and C/EBP has been established in other systems, but this work is a nice demonstration of how it may impact intestinal biology and tumor development.

Point-by-point response to reviewer #2:

Specific points:

1. Human tumor and normal samples were graded for C/EBP α expression, but the actual data (expression grades) are not presented anywhere I could find - only the comparison statistic in Table I. I think it is appropriate to show the scoring - however the authors deem it best to convey the raw data.

Previous data presentation in Figure 1 and Table 1 raised concerns from both reviewers. We have addressed all issues raised by the reviewers to improve the clarity, comprehensibility, visibility of the data presented in the revised Figure1 and a new Supplemental Table1 (see also Point 1 and 2, reviewer #1). In particular, we rephrased the text passages, included a more detailed Supplemental Table 1 that shows the data on C/EBP α expressing areas and the

scoring of the intensity of C/EBP α expression. We also revised Figure 1 as outlined above and incorporated data previously shown in Table 1 now in Figure 1F (decrease of C/EBP α expressing area from healthy to tumor tissue).

2. The authors make the statement: "cells with the most EdU labeling had the lowest C/EBP α expression" to conclude an "inverse relationship between S-phase and C/EBP α expression". However, EdU/BrdU incorporation is more of a binary process - if cells are uptaking the analog they are going through S-phase, less signal may mean they are progressing more slowly, or later in the process, but still in S-phase. Can the author clarify this statement? Perhaps it was meant to imply that C/EBP α positive cells less frequently show EdU uptake?

We thank the reviewer for pointing this out and we rephrased the text accordingly:

"... However, cells labeled with EdU had the lowest C/EBP α expression, implying that C/EBP α positive cells enter S-phase less frequently." (p. 5)

3. On Page 7, the authors state " [organoids] had reduced villi-like areas, and formed larger cells and crypt buds". I can't find any direct data to support this, other than one example image in Figure 4. To make this claim would require further examples and quantitation of crypt bud number, cells size, and differentiated area.

We agree with the reviewer that the morphological changes of organoids were not sufficiently supported by the data shown. Indeed, morphological changes of organoids with stabilized β -Catenin (homozygous) with reduced differentiation were describe before (PalomaOrdóñez-Morán et al 2015, Pamela Riemer et al. 2017, Alexandra L. Farrall et al. 2012). We aimed to use such model (heterozygous) to observe the effect on C/EBP α , as demonstrated in Figure 4. To expand the focus here on the morphological regulation by β -catenin is not within in scope of that manuscript. We decided to keep the focus on the effect of β -catenin on C/EBP α and rephrased the sentence accordingly.

"Organoids with elevated β -catenin exhibited increase in Wnt target gene expression after the induction of recombination, as determined by qRT-PCR for Axin2 and the Wnt-dependent stem cell marker Lgr5 (Figure 4A). C/EBP α expression was severely reduced in GOF β -catenin organoids, as assessed by histological staining (Figure 4B) and after the induction of recombination protein blotting (Figure 4C)." (p.7)

4. The AOM experiment shows some data to support the notion that C/EBP α KOs develop larger tumors, although the data is represented as tumors >5mm. Were the tumors actually measured and can the authors provide the continuous data?

We also measured tumors of other regions of the colon, and of size to up to 0,4cm which showed no significant differences. We provided these data and added them to the Figure 5. In the manuscript we also specified our findings of increased tumor size at the distal part of the colon and adjusted the text accordingly:

"...dysplasias with conditional loss of C/EBP α had significantly increased size in the distal part of the colon, while overall numbers of adenomatous lesions remain unchanged. ..." (p. 8)

Also - please include the AOM/DSS details in the methods section.

We apologize for the neglect (See also Point 4, Reviewer #1). We added the section to the manuscript. (p.14/15)

5. Page 8: "The C/EBP α -depleted colitis associated low grade dysplasia had high nuclear b-catenin levels, although not significantly, as compared to control (Figure 3C)". do the authors mean Figure 5C? And I also cannot find any quantitation of b-catenin levels in this experiment to provide data for this conclusion.

We apologize for leaving out the connection to the refereeing figure, which is now in Supplementary Figure 4A. We now added the reference to Figure S3 that shows the staining for nuclear β -catenin in the DSS/AOM models (see also Point 5, reviewer #1).

"...nuclear β -catenin levels, although not significantly, as compared to the control (Figure 5C, quantification Figure S4C)." (p. 8)

6. Also page 8: The authors refer to "APC target genes". APC is not a transcription factor - I assume this means WNT target genes, but it is important to define this properly.

We tested our expression data with the Sansom_APC_targets gene set (software.broadinstitute.org/gsea/misgdb/cards/SANSOM_APC_TARGETS.html). This gene set shows genes upregulated by loss of APC (adenomatous polyposis coli tumor suppressor protein) that leads to activation of β -catenin/Wnt target genes. We agree with the reviewer that APC target genes is misleading and rephrased the description in the manuscript.

"...Also, Wnt target genes were significantly enriched by testing for a gene set from APC mutant mice (Figure 6D)...." (p. 9)

7. In the list of CRC cell lines, I have never heard of DCD1. Do the authors mean DLD1?

We are thankful for pointing out the typo. The correct term is indeed DLD1. (We corrected this in manuscript and figure).

8. Page 9: "Activation of the *Cebpa* transgene abrogated the clonogenicity of the HCT116...". I disagree with the use of the word 'abrogate'. This implies the effect was profound and almost complete, but in reality I would say clonogenicity was moderately reduced. It is quantified, so stating the absolute effect would be appropriate.

We modified the text according to the reviewer's suggestion.

"Activation of the *Cebpa* transgene reduced the clonogenicity of the HCT116 cells..." (p. 10)

9. Figure 6: The authors claim that the *C/EBPα* KO organoids had "increase crypt bud number and size" but provide no quantitation on this. Absolutely required to make this statement (and from multiple independent organoid isolates - they can vary line to line).

To better address this issue, we i) tracked (by microscopy) the growth of individual organoids over time comparing organoids with similar initial size (see adjusted Figure 6A), and ii) also seeded same cell numbers of organoid fragments and measured increase in total cell numbers after 4 days (see adjusted Figure 6B). The results clearly show the faster growth of *C/EBPα* KO organoids. Accordingly, we changed the manuscript:

"...the homozygous *C/EBPα* KO organoids grew faster, shown by individual tracked organoids over a period of 4 days and measured by the increase in cell number (Figure 6A, B)." (p. 8)

10. Finally, there is one sentence in the discussion/summary that raises multiple concerns for how the data was interpreted. It is: "Our data support the premise that *C/EBPα*: i) controls proliferation and regulated *WNT* target genes, ii) plays a tumor suppressive role in carcinogenesis, and iii) restricts *TA* zone expansion in the gut."

First, I don't think any direct evidence is presented that *C/EBPα* regulates *WNT* target genes. What is present is an inverse correlation between *C/EBPα* and a high *WNT* state. Much more experimental data would be required to say it is directly regulating *WNT* target genes - one easy experiment would be to use the inducible system shown in Figure 7 and determine whether acute *C/EBPα* induction alters *WNT* target genes. A further step would be to show direct binding by ChIP.

We decided to moderate these points, as recommended by the reviewer #2. Indeed, it is very difficult to mechanistically resolve direct repressive functions on the otherwise highly active transcription factor *C/EBPα*. *C/EBPα* repressive mechanisms have previously been explored with somewhat limited mechanistic insight. The best examples suggest repression of *Sox4* (in vivo data; centering on

DNA fragments potentially involved in repression; no mechanistic insight; Tenen lab; Zangh et al. 2013, Cancer Cell, 24:575) and repression of E2F regulated genes by p42 (data published from others and our lab on mechanisms involving E2F; e.g. Slomiany et al, 2000 MCB, 20:5986; Zaragoze et al, 2010, MCB 30:2293; Porse et al., 2001, Cell, 107:247; Kowenz-Leutz et al., BBA, 1859:841). Indeed, it may not be possible to resolve the potential mechanism of WNT-target gene expression by CHIP in a foreseeable time frame, as e.g. weak or indirect binding of C/EBP α might preclude immediate insight (see references, as above). We have therefore decided to change the previous sub-point i) accordingly:

"...i) high C/EBP α and high WNT expression are inversely correlated ..." (p. 10)

However, we evaluated the expression data of C/EBP α KO intestinal organoids for genes with reported Wnt regulatory function in the intestine. Among the downregulated genes in C/EBP α KO organoids we found Ptk6 (protein tyrosine kinase 6) that has been shown to negatively regulate Wnt signaling in the gastrointestinal tract. Ptk6 expression is reduced in human adenocarcinoma and reduction of Ptk6 also promotes growth of xenografts (Mathur et al., 2016). In response to DNA damage Ptk6 is expressed in intestinal crypts and promotes apoptosis by inhibiting prosurvival signaling (Haegebarth et al., 2009). Ptk6 phosphorylates Cdc73 (parafibromin, a component of the RNA Polymerase II associated Paf1C complex) to negatively regulate β -catenin/TCF transcription (Kikuchi et al., 2016; Palka-Hamblin et al., 2010; Shi et al., 1997). In conclusion, C/EBP α might attenuate Wnt/ β -catenin signaling and impact on cancer cell proliferation by controlling expression of Ptk6.

Second, "restrict TA zone expansion in the gut". Is this true outside of organoids (which were not quantified)? If so, I would imagine that the Villin-Cre/CEBPaFL/FL mice would show and expanded TA zone in the normal intestine/colon. Do the authors have any data to support this? I don't think these issues require extensive experimental data, but I don't think the current language is supported by the data.

Our data on C/EBP α KO organoids show clear changes in growth (see also point# 9), ii) during in vivo oncogenic conditions the loss of C/EBP α promotes tumor growth and iii) in CRC cells expression of C/EBP α reduced growth. We did not observe strong effects in vivo under normal conditions. To clarify the issue we rephrased the sentence to:

"...iii) reduces oncogene dependent growth." (p. 10)

December 10, 2018

RE: Life Science Alliance Manuscript #LSA-2018-00173-TR

Prof. Achim Leutz
Max Delbrück Center for Molecular Medicine
Tumorigenesis and Cell Differentiation
Robert-Rössle-Strasse 10
Berlin, Berlin 13125
Germany

Dear Dr. Leutz,

Thank you for submitting your revised manuscript entitled "A C/EBP α -Wnt connection in gut homeostasis and carcinogenesis". We would be happy to publish your paper in Life Science Alliance pending final revisions to address reviewer #1's remaining concern (text changes needed) and to meet our formatting guidelines:

Please deposit your RNA-seq data in a repository (GEO) and add the identifier in a 'data availability' section in the methods part of your paper. Please add scale bars to figures 1, 2, 5C, 6A, S3, and S4. Please add the statistical test used to derive p values in figures 4 and 7.

A. FINAL FILES:

-- High-resolution figure, supplementary figure and video files uploaded as individual files: See our detailed guidelines for preparing your production-ready images, <http://life-science-alliance.org/authorguide>

B. MANUSCRIPT ORGANIZATION AND FORMATTING:

Full guidelines are available on our Instructions for Authors page, <http://life-science-alliance.org/authorguide>

Sincerely,

Reviewer #1 (Comments to the Authors (Required)):

The authors have dealt well with the critical remarks from my side. In particular, the new figure 1 presents human tumor data in a much clearer way than in the previous version, revealing a seeming focal downregulation of C/EBPa in colorectal tumors compared to normal tissue. However, the limited number of cases do not allow the statement on page 11 that "reduction of areas expressing

C/EBPa were ... predictive", if this should mean that they are of prognostic value. Except of two outliers with extremely reduced C/EBPa expression adenomas and adenocarcinomas were very similar. It therefore suffices to state that adenomas and adenocarcinomas showed areas of absence of C/EBPa expression in most cases. In that respect, it is a bit worrying that there is a great variation of C/EBPa intensity even between normal tissues, with 9 out of 17 samples showing weak expression, leading to an average of the expression scoring of 1.4, similar to the average of the adenocarcinoma samples. Although the trend of focal absence of C/EBPa expression in tumors is clear, this variability limits somewhat the validity of the main conclusion of Figure 1. The authors should acknowledge that issue when discussing their data.

Reviewer #2 (Comments to the Authors (Required)):

The authors have addressed all of my (already) minor comments and amended the language and interpretation/description of their data where appropriate.

December 18, 2018

RE: Life Science Alliance Manuscript #LSA-2018-00173-TRR

Prof. Achim Leutz
Max Delbrück Center for Molecular Medicine
Tumorigenesis and Cell Differentiation
Robert-Rössle-Strasse 10
Berlin, Berlin 13125
Germany

Dear Dr. Leutz,

Thank you for submitting your Research Article entitled "A C/EBP α -Wnt connection in gut homeostasis and carcinogenesis". It is a pleasure to let you know that your manuscript is now accepted for publication in Life Science Alliance. Congratulations on this interesting work.

*****IMPORTANT:** If you will be unreachable at any time, please provide us with the email address of an alternate author. Failure to respond to routine queries may lead to unavoidable delays in publication.*******

DISTRIBUTION OF MATERIALS:

Again, congratulations on a very nice paper. I hope you found the review process to be constructive and are pleased with how the manuscript was handled editorially. We look forward to future exciting submissions from your lab.

Sincerely,
